# The potential of X-ray computed tomography for xylological and dendrochronological analyses of Egyptian mummy labels

**François Blondel** [1,2]*, **Gisela Bélot**[3], **Christophe Corona**[1,4], **Sabine R. Huebner**[5], **Markus Stoffel**[1,6,7]

1 Climate Change Impacts and Risks in the Anthropocene (C-CIA) Institute for Environmental Sciences University of Geneva, Geneva, Switzerland, 2 Laboratory Chrono-environnement, UMR 6249, Besançon, France, 3 Bibliothèque Nationale et Universitaire (BNU), Strasbourg, France, 4 Laboratory of Alpine Ecology (LECA), UMR 5553, Gières, France, 5 Department of Ancient Civilizations, Institute of Ancient History, University of Basel, Basel, Switzerland, 6 Dendrolab.ch, Department of Earth Sciences, University of Geneva, Geneva, Switzerland, 7 Department F.-A. Forel for Environmental and Aquatic Sciences, University of Geneva, Geneva, Switzerland

* francois.blondel@unige.ch

**Data Availability Statement:** The data underlying the results presented in the study are available from: Zenodo https://zenodo.org/records/10376111 https://zenodo.org/records/10384262.

## Abstract

X-ray computed tomography (XRCT) imaging allows non-destructive visualization of the structure of various materials. Applied to wooden objects, it allows determination of their morphologies or manufacturing techniques, but also measurement of growth ring widths. We have applied XRCT to a selection of 38 mummy labels. This funerary furniture, made up of endemic or imported tree species, has survived thanks to environmental conditions in very large quantities in regions in Middle and Upper Egypt and is featured now in museum collections across the globe. Mummy labels thus represent a unique and abundant data source to build floating or absolutely dated dendrochronological chronologies for this period. Here we discuss the possible contributions and limitations of XRCT for the analysis of these artifacts and show that the approach allows identification of discriminating markers for the identification of certain species on the transverse plane, but that the insufficient resolution of the tangential and radial planes normally prevents formal identification of species. By contrast, XRCT undeniably enhances the visibility of toolmarks (in terms of numbers and depth), and thereby allows highlighting marks that remain invisible to the naked eye; XRCT also provides key insights into cutting methods and the calibers used and yields new information on silvicultural practices and the knowhow of Egyptian craftsmen. Finally, the measurement of ring widths on XRCT imagery is also more accurate than what can be achieved by traditional dendrochronological measurements, especially in the case of cuts realized on a slab. The approach also confirms the limited potential of local broadleaved species for dendrochronological approaches due to unreadable or poorly visible tree rings and mostly short tree-ring sequences.

## Introduction

Dendrochronological analyses are sometimes performed on wooden objects from museum collections [1]: these may include architectural wood (e.g., boats, construction wood) [2, 3],

**Funding:** Yes. This article is part of a post-doctoral project funded by the SNSF, project n°192176 "The Roman Egypt Laboratory: Climat Change, Societal Transformations, and the Transition to Late Antiquity" web page project: https://ancientclimate. philhist.unibas.ch/en/project/ Web page SNSF: https://www.snf.ch/fr The funders did not play any role in the study design, data collection or analysis, neither did they influence the decision to prepare or publish the manuscript.

**Competing interests:** NO authors have competing interests.

domestic objects [4, 5] or pieces of art [6, 7]. Analyses on objects conserved in museums are performed on woods from the Neolithic to the Common Era. They necessarily have to rely, however, whenever possible, on non-destructive approaches [1] such as high-resolution digital photography, hyper-spectral imaging, fast ion beam analysis [8] or X-ray computed tomography (XRCT). The latter allows visualization of the internal structure of wood without affecting its integrity [4, 7, 9–11]. It allows an internal view of the anatomy of the species used, but also provides archaeological (i.e. cutting and collection methods, calibers exploited) and socio-environmental (e.g., silvicultural practices) information [12–14]. At the same time, XRCT also allows highlighting the dendrochronological potential of tree species [9]. The latter depends in particular on the presence of discernible rings boundaries, but also on the annual character of growth rings. In this study, XRCT was tested on a batch of mummy labels from Roman Egypt. Mainly dated from the 1st to the 4th centuries, these wooden objects can provide information on the shaping techniques used, the selection of wood species for funerary purposes and the position of cuts within the trees used [15]. Middle and Upper Egypt, located at the interface between inhabited territories and the desert, are auspicious regions for the discovery and study of "organic archives" (e.g., papyrus, wood, leather) thanks to their arid environment in which objects have been conserved over millennia. The objective of this article is to assess and describe the potential of XRCT to study the wood anatomy, toolmarks and dendrochronological potential of 38 mummy labels at the National and University Library (BNU) of Strasbourg. Due to the origin of the wood used, assessing the dendrochronological potential of the available species would eventually allow the construction of reference chronologies from the Eastern Mediterranean for the Roman period, for which only floating chronologies exist for the time being [16–19]. Yet, the creation of such reference chronologies remains challenging given the importance of wood reuse in Egyptian craftsmanship [28].

## Materials and method

### Mummy labels from the BNU in Strasbourg

The BNU of Strasbourg has 256 mummy labels and various documentary wooden labels [20]. The collection was assembled between 1895 and 1913 by the Egyptologist Wilhelm Spiegelberg, and contains mainly labels from Sohag and Akhmim in Egypt dating to the Greco-Roman period (i.e. from 323 to 30 BC for the Ptolemaic period and from 39 BC to 330 AD for the Roman period).

Spiegelberg acquired 206 mummy labels (HO20-HO225; HO256) from Robert Forrer (archaeologist, writer, collector) after 1893 and made further acquisitions of labels inscribed in demotic and Greek-Demotic between 1895 and 1913. The latter include tax receipts and accounting records from Gebelein (Pathyris) and Akhmim (Panopolis) as well as 8 labels inscribed in hieratic. The Strasbourg mummy label also was an important source for Spiegelberg's onomastic study of the *Ägyptische und Griechische Eigennamen* [21]. Many Egyptologists, including F. Preisigke (between 1913 and 1922) and U. Kaplony-Heckel (in 1966) contributed to the regular edition of the texts of the collection, but it was necessary to wait for the publication of S. P. Vleeming to have a real compendium of the edited texts of the Demotic and Greek labels, including the collection at BNU Strasbourg [22].

These labels, most often made of wood, were pierced at one end to be attached to the embalmed mummies with a thread when the latter were moved from the place of mummification to the place of burial [23] (Fig 1). The great diversity of shapes, materials (wood, stone, terracotta, enameled glass), wood species used and inscriptions do not always allow identification of the origin of the manufacturers (mummification workshops, families or wood craftsmen) [15]. Most often, however, the labels contain basic information about the deceased: name and

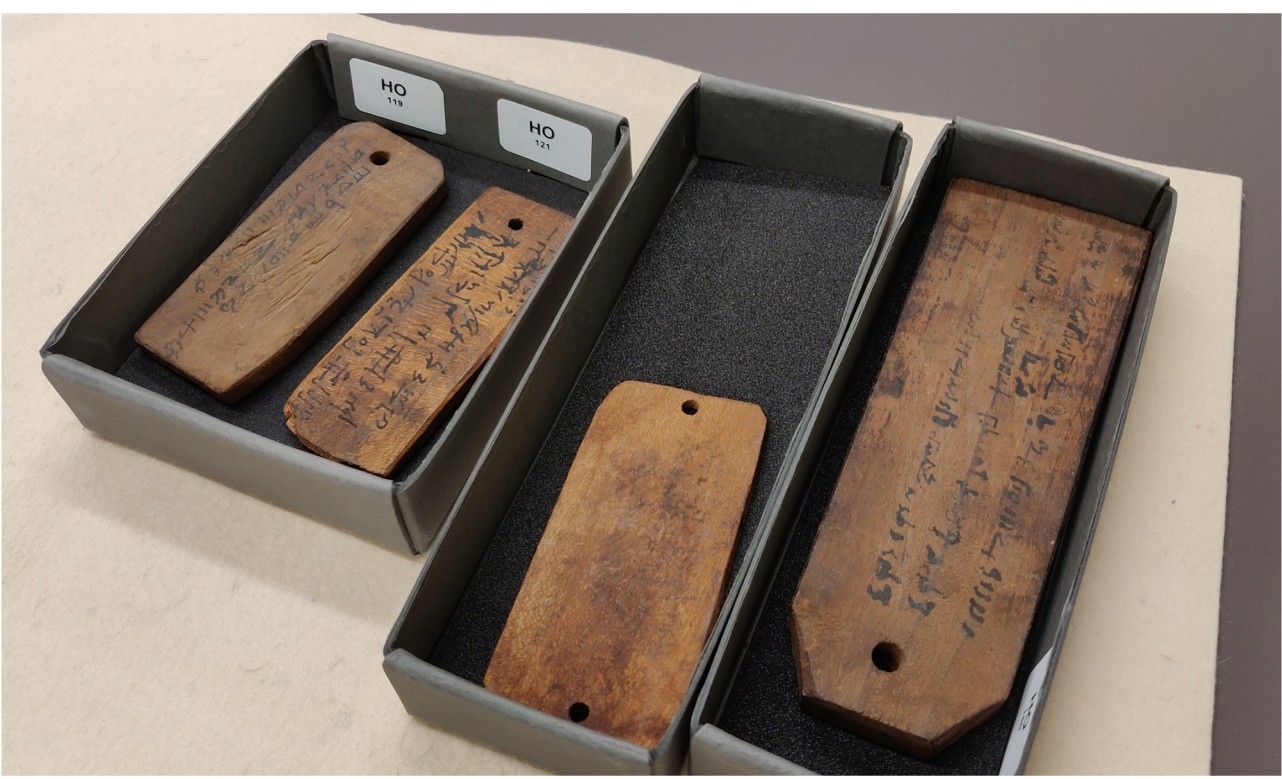

**Fig 1. Examples of mummy labels from the collection of the BNU of Strasbourg.**

origin (filiation), age and profession, sometimes a name of a town or region, the mummification method, or a burial destination [23]. The text is in Greek or Demotic, often both. It may be written in black ink (more rarely red), or engraved [15, 20]. These labels served two main functions, that is identification of the body until burial as well as religious and protective functions to accompany the deceased to the afterlife.

The labels vary greatly in shape: they were classified using the typology proposed by Blondel *et al*. [15], which in turn was inspired by Quaegebeur [24], Gaudard *et al*. [23] and Worp [25]. They are sometimes made from recycled wood as attested by toolmarks (i.e. remnant of a groove, dowel hole, nail hole) that are unrelated to the purpose of the label [15]. Labels are most often sawn, and then, for the most part, flattened to be inscribed or engraved. Some clumsiness in writing repeatedly attests to the limited expertise in funerary establishments, woodworking shops, or families. Other labels show a perfect mastery of woodworking, writing, and even drawing (e.g., representations of religious symbols). The latter labels do not come from the most disadvantaged social strata because they also belonged to deceased who underwent an expensive mummification process [26].

## X-ray computer tomography

An EasyTom 150/160 X-ray tomograph (RX Solutions) was used in this study on a selection of mummy labels from the BNU as it can scan objects up to 380 mm in height and 180 mm in diameter. Placed on a support that guarantees protection and stability, each label was placed on a rotating plate located between the X-ray source and the 2D detector (Fig 2) and then X-rayed at 360˚. The scanner parameters for the session carried out on the mummy labels were set at 90 Kv with an intensity of 195 mA for an acquisition resolution varying between 11 and

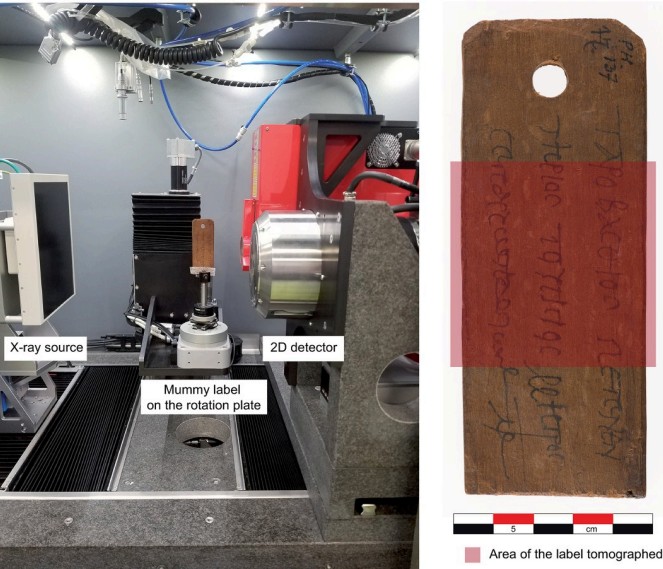

**Fig 2. Location of a label (HO137) on its support/vice in the tomograph between the X-ray source and the 2D detector and detail of the tomographed area on the label (Coll. and photogr. BNU de Strasbourg).**

42 μm with 2016 projections (that is about 20 images on average per projection) with a frame rate of 12,5 and a temperature of 28˚C. Each image was reconstructed by filtered retroprojection using the XAct software (RX Solutions). The acquired images were then visualized and processed with the VGStudio Max software (VolumeGraphics) to restore a 3D view of the object. The tomograph and software also allowed working in 2D on the transverse, tangential and radial planes to visualize the internal structure of the wood and to highlight its anatomical characteristics without physical cutting.

## Selection of tomographed labels

The selection of tomographed mummy labels was based on their anatomical (i.e. taxonomic) diversity (by distinguishing between local broadleaved, imported and conifer using binocular observation of the transverse plane) or their dendrochronological potential (i.e. number of rings present to perform dating). We selected 38 labels from a body of imported coniferous (n = 11), imported broadleaved (n = 10) and local (n = 17) species. Tomographic acquisitions were taken on a small portion of the label to obtain a complete dendrochronological sequence at very high resolution. The resolution varied according to the width (32–95 mm) and thickness (5–16 mm) of the labels due to the absorption of the X-ray beam. The detailed description of each of the selected labels is provided in Table 1. Dated mainly on the basis of style, inscription, and place of provenance, they cover a period from the 1[st] to the 4[th] century CE and originate for the most part (27 out of 38) from the necropolis of Sohag (n = 21), Bompae (n = 3), and Panopolis (n = 3) located in Upper Egypt. A vast majority of the labels (n = 27) are rectangular in shape, nine are trapezoidal, one is in the shape of a stele with a handle and one is in the shape of a *Tabula Ansata* (see Blondel *et al.*, 2023 [15] for details on mummy label styles and shapes).

## Anatomical identifications

Due to the insufficient resolution of XRCT for the analysis of wood structures along the tangential and radial planes [9], we limited analyses to the transverse plane as it allows for optimal

**Table 1. Inventory of mummy labels selected for XRCT with key descriptive information and regarding CT resolution.**

| Museum inventory | First Edition | Period | Provenance | Format | Tree genus | Dimensions (in cm) | | | Resolution | No. of projections |
|---|---|---|---|---|---|---|---|---|---|---|
| | | | | | | Length | width | Thickness | | |
| Ho / PH / T. 018 | Short Texts 1 496 | Unknown | Unknown | Rectangular | local broadleaves? | 14,4 | 4,1 | 0,9 | 30 μm | 1293 |
| Ho / PH / T. 028 | Short Texts 2 822, SB 1 5441, C.Étiq.Mom. 1730 | 225–275 CE | Sohag | Rectangular | local broadleaves? | 12,1 | 6,45 | 0,95 | 40 μm | 1285 |
| Ho / PH / T. 033 | SB 1 5398, T.Spiegelberg 82, C. Étiq.Mom. 421 | I—IV CE | Sohag | Rectangular | imported conifers | 11,5 | 4,5 | 0,8 | 27 μm | 1259 |
| Ho / PH / T. 041 | Short Texts 2 756, SB 1 5448, C.Étiq.Mom. 1731 | 225–275 CE | Sohag | Rectangular | local broadleaves | 13,2 | 5,95 | 0,65 | 36 μm | 1206 |
| Ho / PH / T. 042 | SB 1 5449, C.Étiq.Mom. 1732 | 100–299 CE | Sohag | Rectangular | imported broadleaves? | 11,85 | 5,1 | 0,65 | 31 μm | 1265 |
| Ho / PH / T. 043 | Short Texts 2 703, SB 1 5450, C.Étiq.Mom. 1733 | 200–225 CE | Sohag | Rectangular | imported conifers | 13,2 | 5,6 | 0,8 | 35 μm | 1264 |
| Ho / PH / T. 047 | SB 1 5396, T.Spiegelberg 80, C. Étiq.Mom. 419 | I—IV CE | Sohag | Rectangular | local broadleaves | 11,9 | 5,9 | 1,05 | 35 μm | 1279 |
| Ho / PH / T. 058 | SB 1 5372, T. Spiegelberg, 55 | Unknown | Unknown | Rectangular | local broadleaves | 12,4 | 5,35 | 0,8 | 32 μm | 1279 |
| Ho / PH / T. 061 | SB 1 5455, C.Étiq.Mom. 459, T. Spiegelberg 95 | Unknown | Unknown | Rectangular | local broadleaves? | 12,25 | 5,4 | 0,8 | 32 μm | 1302 |
| Ho / PH / T. 065 | SB 1 5458, C.Étiq.Mom. 461 | Unknown | Unknown | Trapezoidal | imported broadleaves? | 12,2 | 5,3 | 1,15 | 31 μm | 1279 |
| Ho / PH / T. 068 | SB 1 5389, T.Spiegelberg 73, C. Étiq.Mom. 414 | I—IV CE | Sohag | Rectangular | imported broadleaves? | 15,5 | 7,1 | 0,8 | 40 μm | 1279 |
| Ho / PH / T. 069 | SB 1 5393, T.Spiegelberg 77, C. Étiq.Mom. 417 | I—IV CE | Sohag | Rectangular | local broadleaves | 14,85 | 5,5 | 0,8 | 30 μm | 1279 |
| Ho / PH / T. 085 | SB 1 5373, T.Spiegelberg 56, C. Étiq.Mom. 401 | I—IV CE | Unknown | Rectangular | imported conifers | 11,5 | 5,5 | 0,8 | 35 μm | 1279 |
| Ho / PH / T. 095 | SB 1 5470, C.Étiq.Mom. 1903 | 100–299 CE | Sohag | Rectangular | local broadleaves | 14,4 | 5,65 | 1,35 | 34 μm | 1279 |
| Ho / PH / T. 097 | SB 1 5472, C.Étiq.Mom. 1742 | 100–299 CE | Sohag | Rectangular | local broadleaves | 11,2 | 5 | 1,05 | 32 μm | 1265 |
| Ho / PH / T. 104 | SB 1 4192, T.Spiegelberg 14, C. Étiq.Mom. 1707, Short Texts 2 824 | 200–299 CE | Bompae | Rectangular | local broadleaves | 12,2 | 6,4 | 0,95 | 37 μm | 1279 |
| Ho / PH / T. 105 | SB 1 5397, T.Spiegelberg 81, C. Étiq.Mom. 420 | I—IV CE | Sohag | Rectangular | local broadleaves | 11,25 | 4,35 | 1,25 | 26 μm | 1268 |
| Ho / PH / T. 106 | Short texts 1 594 | Unknown | Unknown | Rectangular | imported conifers | 13,7 | 4,1 | 1,15 | 15 μm | 1048 |
| Ho / PH / T. 108 | SB 1 5417, T.Spiegelberg 101, C.Étiq.Mom. 436 | I—IV CE | Sohag | Trapezoidal | imported conifers | 11,95 | 6,35 | 1,2 | 20 μm | 960 |
| Ho / PH / T. 111 | SB 1 4186, T.Spiegelberg 7, C. Étiq.Mom. 343, Short Texts 2 637 | 100–299 CE | Bompae | Rectangular | local broadleaves? | 16,1 | 4,8 | 0,75 | 30 μm | 1279 |
| Ho / PH / T. 112 | SB 1 5385, T.Spiegelberg 69, C. Étiq.Mom. 411 | I—IV CE | Sohag | Trapezoidal | local broadleaves? | 14,4 | 6,5 | 1,05 | 36 μm | 1279 |
| Ho / PH / T. 117 | SB 1 5480, SB 1 5410, C.Étiq. Mom. 472, T.Spiegelberg 94 | I—IV CE | Sohag | Trapezoidal | local broadleaves? | 10,4 | 6,45 | 1 | 38 μm | 1231 |
| Ho / PH / T. 132 | SB 1 5487, C.Étiq.Mom 1926 | 100–299 CE | Sohag | Trapezoidal | local broadleaves | 14,55 | 6,5 | 1,35 | 38 μm | 1263 |
| Ho / PH / T. 137 | Short Texts 2 664, SB 1 5489, C.Étiq.Mom. 478 | 100–299 CE | Sohag | Rectangular | imported broadleaves | 11,4 | 4,4 | 0,5 | 32 μm | 1646 |
| Ho / PH / T. 138 | Short Texts 2 794, SB 1 5490, C.Étiq.Mom. 479 | 200–299 CE | Sohag | Trapezoidal | imported broadleaves | 10,45 | 5 | 0,85 | 31 μm | 1279 |

*(Continued)*

**Table 1.** (Continued)

| Museum inventory | First Edition | Period | Provenance | Format | Tree genus | Dimensions (in cm) | | | Resolution | No. of projections |
|---|---|---|---|---|---|---|---|---|---|---|
| | | | | | | Length | width | Thickness | | |
| Ho / PH / T. 141 | SB 1 5492, C.Étiq.Mom. 481 | 100–299 CE | Sohag | Rectangular | imported broadleaves? | 11,3 | 4,8 | 0,65 | 30 μm | 1279 |
| Ho / PH / T. 162 | SB 1 5502, C.Étiq.Mom. 488 | 100–299 CE | Sohag | Rectangular | local broadleaves | 17,7 | 5,7 | 1,4 | 34 μm | 1266 |
| Ho / PH / T. 170 | Short Texts 2 683, SB 1 5506, C.Étiq.Mom. 1753 | 200–225 CE | Sohag | Trapezoidal | imported conifers | 14,7 | 5,8 | 1,05 | 33 μm | 1279 |
| Ho / PH / T. 172 | SB 1 4194, T.Spiegelberg 17, SB 1 5507, C.Étiq.Mom. 348, C.Étiq.Mom. 490, Short Texts 2 757 | 220–299 CE | Panopolis (?) | Rectangular | imported conifers | 13,5 | 6,9 | 1,2 | 42 μm | 1279 |
| Ho / PH / T. 182 | SB 1 4200, T.Spiegelberg 23, C.Étiq.Mom. 1709, Short Texts 2 684 | 200–250 CE | Bompae | Rectangular | imported conifers | 11,9 | 4,8 | 0,9 | 30 μm | 1279 |
| Ho / PH / T. 184 | SB 1 5511, C.Étiq.Mom. 1754, Short Text 2 750 | 100–299 CE | Panopolis/ Bompae (?) | Rectangular | imported conifers | 11,9 | 5,2 | 1,1 | 32 μm | 1269 |
| Ho / PH / T. 199 | T.Spiegelberg 41, Short Texts 2 574 | III CE | Panopolis (?) | Trapezoidal | imported broadleaves | 8,3 | 3,2 | 0,5 | 11 μm | 1079 |
| Ho / PH / T. 210 | SB 1 5525, C.Étiq.Mom. 502 | Unknown | Unknown | Stela-shape with handle | imported broadleaves | 12,55 | 8 | 0,85 | 28 μm | 1112 |
| Ho / PH / T. 211 | SB 1 5526, C.Étiq.Mom. 1978, T.Spiegelberg 107, SB 1 5423 descr. | I—IV CE | Sohag | Trapezoidal | imported conifers | 12,8 | 6,1 | 1 | 35 μm | 1310 |
| Ho / PH / T. 225 | SB 1 5536, C.Étiq.Mom. 510, Short Texts 2 741 | 100–299 CE | Panopolis (?) | Rectangular | imported conifers | 10,3 | 5 | 0,85 | 30 μm | 1248 |
| Ho / PH / T. 226 | SB 1 5537, C.Étiq.Mom. 2122 | Unknown | Unknown | Tabula ansata | local broadleaves | 23,55 | 9,5 | 1,2 | 32 μm | 1062 |
| Ho / PH / T. 229 | SB 1 5538, C.Étiq.Mom. 1936 | Roman | Unknown | Rectangular | imported broadleaves | 15,75 | 9,25 | 1,6 | 31 μm | 1140 |
| Ho / PH / T. 255 | Unknown | Unknown | Unknown | Rectangular | local broadleaves? | 11,4 | 7,7 | 0,6 | 26 μm | 949 |

visualization of growth rings. In the case of broadleaved species, anatomical markers included pore numbers shape, and distribution as well as the distinction between earlywood and latewood, ring boundaries, the presence or absence of parenchyma, as well as ray widths. In the case of conifer species, we limited analyses to the transition from earlywood to latewood, the presence of resin ducts and signs of traumatic cell structures.

On the basis of these anatomical markers, we propose several hypotheses concerning the species used. These hypotheses are then validated against (1) different species identified on several types of Roman Egyptian furniture (e.g., mummy labels, mummy portraits, everyday objects; [27–30]; as well as (2) wood anatomy and macroscopic wood characteristics (John [Unpublished]) [31–35].

## Size estimation and technological observations

The estimation of the caliber of the processed wood allows highlighting possible selection criteria of a tree for the cutting of the mummy labels. The estimation of the distance to the pith was performed from the convergence of the radii and the curvature of the rings from the cross-sectional planes obtained with XRCT [12, 36–38] (Fig 3). Because tree growth is not perfectly concentric, the distance to the pith is calculated as an average between possible minimum and

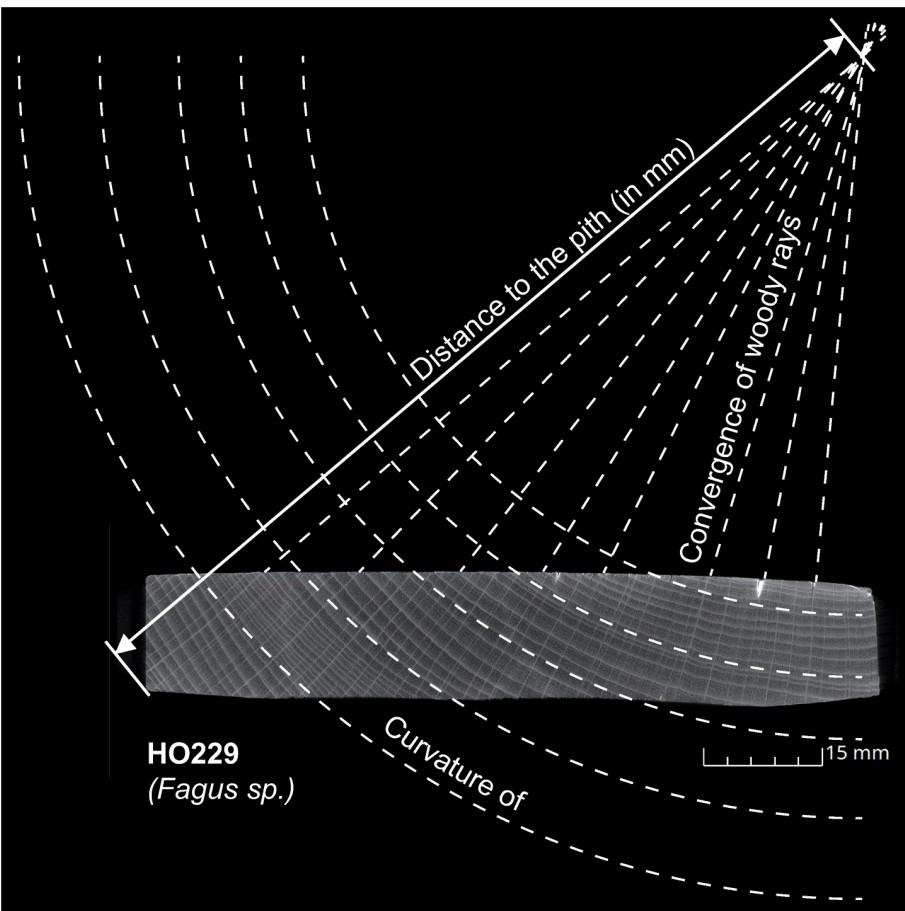

**Fig 3. Methodology applied for the estimation of the size of trees selected for the cutting of mummy labels from the transverse plane of label HO229.**

maximum values. Tree age could not always be estimated due to poorly visible ring boundaries in some species.

The cutting methods can sometimes be identified by the naked eye on the ends of objects. However, toolmarks or secretions sometimes limit these observations. We used XRCT to extend these observations to all tomographed labels. Finally, the 3D model obtained from the XRCT images was also used to highlight toolmarks. These were compared to the visual observations made on the labels to evaluate a possible added value of XRCT images.

## Ring width measurements

On labels made of conifer wood, the ring widths could be measured conventionally because of the distinct ring boundaries visible on the label's surface. Yet, we measured ring widths only on labels containing more than 30 rings. Measurements (0.01 mm) were made using (1) a Lintab 5 measuring table equipped with a binocular magnifier and Tsap Win v. 4.82b2 software [39] or (2) from high-resolution photographs (300 dpi minimum) using CooRecorder software [40]. To prevent errors related to illegible, partial or false rings [41], measurements were made on the flat or on both sides of each object and along several transects (Fig 4). The different series were then synchronized using Tsap Win software. These ring series were compared to

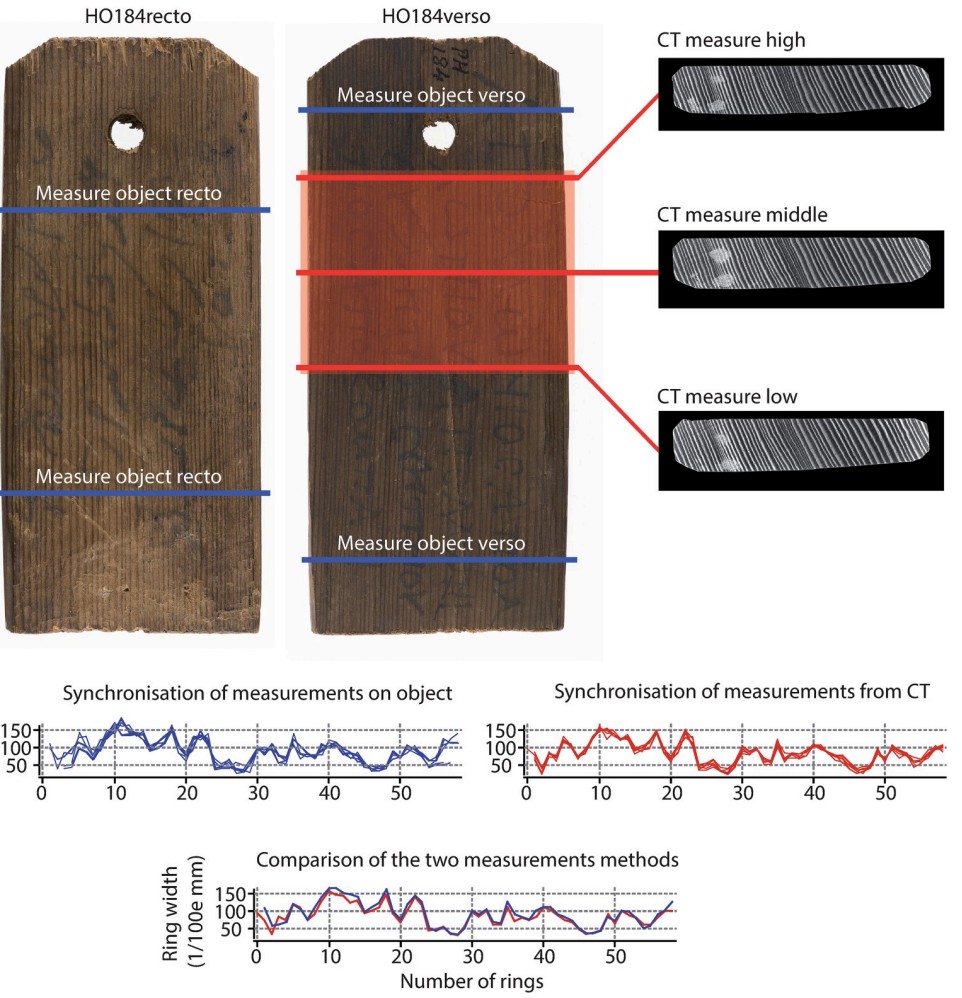

**Fig 4. Presentation of the two methods of measuring ring widths on label HO184: One directly on the object and the other from XRCT images, as well as the comparison of the ring widths of the two synchronizations obtained (coll. and photogr. BNU de Strasbourg HO184).**

the series made on the transverse plane of the XRCT images. The latter were made and synchronized in the same way, on at least two or three transects using the CooRecorder and TSAPWIN software.

## Results

### Identification of the species used

Using the anatomical criteria visible in the transverse plane of XRCT images, we can distinguish imported broadleaved species from those endemic to Egypt as well as from imported conifer species. We were also able to hypothesize the exact species used for 27 out of the 38 labels (Fig 5). Although the anatomy of labels HO18 and HO255 appears to be similar, we could not identify the species in these cases.

For the imported broadleaved species, we hypothesize that labels HO137, 199, 210 and 229 are made from beech (*Fagus* sp.). This hypothesis is linked to the presence of diffuse pores, a semi-porous zone, and the presence of uniserial or multi-serial woody rays that thicken at the

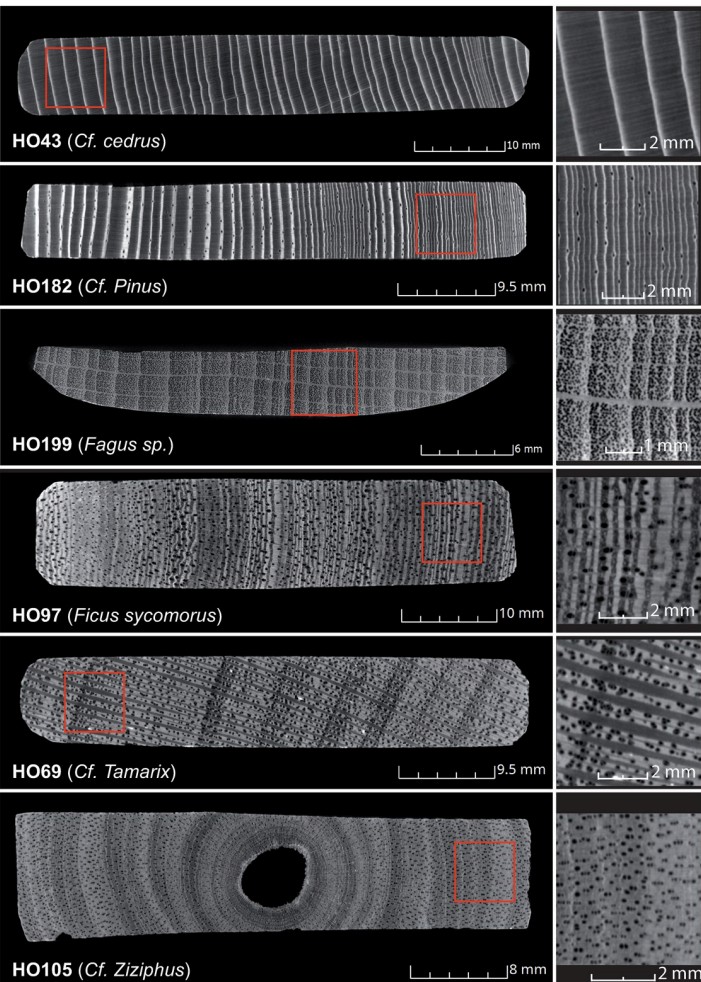

**Fig 5. Tomographic images of the transverse plane of six mummy labels made of different endemic and imported species.** Detailed view of the wood anatomy, highlighting the limits of the resolution of XRCT resolutions.

edge of rings, with the latter being perfectly legible. Similarly, using the criteria described in Table 2, labels HO42 and 141 appear to be made of willow (*Salix* sp.) wood while label HO138 is potentially made from elm (*Ulmus* sp.) wood.

In the case of the endemic broadleaved species, seven labels (HO28, 41, 69, 95, 104, 132, and 226) show (1) groups of two to three diffuse pores, sometimes arranged radially, (2) an earlywood with a semi-porous zone, and (3) broad woody rays, sometimes widening at the ring boundaries. Based on these characteristics, they were probably made of tamarisk (*Tamarix* sp.) wood. Using further discriminating criteria (Table 2), labels HO47, 58, and 97 are likely made of fig (*Ficus sycomores*) wood, whereas labels HO111 and 162 are likely acacia (*Acacia* sp.), although a distinction between the two species is sometimes complex when relying on the transverse plane alone. Last but not least, label HO105 seems to be made of jujube (*Ziziphus spina-christi*) wood.

For conifer species, discriminating markers are poorly present on the transversal plane, rendering species determination complex (Table 2). Five labels (HO33, 170, 182, 184 and 211) are potentially made of pine (*Pinus* sp.) because of the presence of resin ducts in the latewood and a fairly abrupt transition from early- to latewood. Two labels (HO85 and 172) without resin

**Table 2. Detailed description of the different anatomical markers identified on the transverse plane of the XRCT images and–to a lesser extent–based on observations made on the mummy labels.**

| Museum inv. num. | Pore arrangement | Rays width | Limit of tree-rings | Other anatomical markers | Hypothesis species |
|---|---|---|---|---|---|
| **Imported hardwood** | | | | | |
| Ho / PH / T. 42 | Diffuse pores, radially bonded with a slightly semi-porous area | Uniseriate | Distinct | Groups of 2 to 3 pores radially | *Salix sp.* |
| Ho / PH / T. 65 | Diffuse pores (small pores), sometimes with a slightly porous area in the initial wood | Uniseriate? | More or less distinct | | Undetermined |
| Ho / PH / T. 68 | Diffuse pores (small pores), sometimes with a slightly porous area in the initial wood | Uniseriate? | More or less distinct | | Undetermined |
| Ho / PH / T. 112 | Diffuse pores sometimes clustered together or radially, with a semi-porous zone | Uniseriate or 2 seriate? | More or less distinct | | Undetermined |
| Ho / PH / T. 137 | Diffuse pores with a semi-porous zone | Uni- and multiseriate | Distinct | Wider woody rays at the boundary of the ring | *Fagus sp.* |
| Ho / PH / T. 138 | Porous zone composed of a single series of pores | Wide rays | Distinct | Several bands of tangential pores, sometimes slightly oblique with parenchyma | *Ulmus sp. Minor* |
| Ho / PH / T. 141 | Diffuse pores with a semi-porous zone | Uniseriate? | More or less distinct | | *Cf. salix* |
| Ho / PH / T. 199 | Diffuse pores with a semi-porous zone | Uni- and multiseriate | Distinct | Wider woody rays at the boundary of the ring | *Fagus sp.* |
| Ho / PH / T. 210 | Diffuse pores with a semi-porous zone | Uni- and multiseriate | Distinct | Wider woody rays at the boundary of the ring | *Fagus sp.* |
| Ho / PH / T. 229 | Diffuse pores with a semi-porous zone | Uni- and multiseriate | Distinct | Wider woody rays at the boundary of the ring | *Fagus sp.* |
| **Local hardwood** | | | | | |
| Ho / PH / T. 18 | Diffuse pores, grouped by two or three | 2 seriate or more | More or less distinct | Numerous xylophagous galleries | Undetermined |
| Ho / PH / T. 28 | Diffuse pores, grouped from two to three pores and/or radially, sometimes with a semi-porous zone | Wide to very wide | More or less distinct | Expansion of the radii at the boundary of the ring | *Tamarix sp.* |
| Ho / PH / T. 41 | Diffuse pores, grouped from two to three pores and/or radially, sometimes with a semi-porous zone | Wide to very wide | More or less distinct | Expansion of the radii at the boundary of the ring | *Tamarix sp.* |
| Ho / PH / T. 47 | Diffuse pores, with group of pores by two or three sometimes arranged radially | Wide rays | unreadable | Alternating parenchyma bands and fiber | *Ficus sp.* |
| Ho / PH / T. 58 | Diffuse pores, with group of pores by two or three sometimes arranged radially | Wide rays | unreadable | Alternating parenchyma bands and fiber | *Ficus sp.* |
| Ho / PH / T. 61 | Diffuse pores (small pores) not very close together | Uniseriate? | unreadable | | Undetermined |
| Ho / PH / T. 69 | Diffuse pores, grouped from two to three pores and/or radially, sometimes with a semi-porous zone | Wide to very wide | More or less distinct | Expansion of the radii at the boundary of the ring | *Tamarix sp.* |
| Ho / PH / T. 95 | Diffuse pores, grouped from two to three pores and/or radially, sometimes with a semi-porous zone | Wide to very wide | More or less distinct | Expansion of the radii at the boundary of the ring | *Tamarix sp.* |
| Ho / PH / T. 97 | Diffuse pores, with group of pores by two or three sometimes arranged radially | Wide rays | unreadable | Alternating parenchyma bands and fiber | *Ficus sp.* |
| Ho / PH / T. 104 | Diffuse pores, grouped from two to three pores and/or radially, sometimes with a semi-porous zone | Wide to very wide | More or less distinct | Expansion of the radii at the boundary of the ring | *Tamarix sp.* |
| Ho / PH / T. 105 | Diffuse pores, with group of pores by two to four arranged radially, rarely isolated | Uniseriate? | More or less distinct | Thin band of parenchyma | *Ziziphus sp.* |
| Ho / PH / T. 111 | Diffuse pores, with groups of pores by two or in clusters and sometimes presenting a semi-porous zone | Wide rays | unreadable | Alternating parenchyma bands and fiber | *Cf. Ficus ou acacia* |

*(Continued)*

**Table 2.** (Continued)

| Ho / PH / T. 117 | Diffuse pores | Uniseriate | unreadable | | Undetermined |
|---|---|---|---|---|---|
| Ho / PH / T. 132 | Diffuse pores, grouped from two to three pores and/or radially, sometimes with a semi-porous zone | Wide to very wide | More or less distinct | Expansion of the radii at the boundary of the ring | *Tamarix sp.* |
| Ho / PH / T. 162 | Diffuse pores, with groups of pores by two or in clusters and sometimes presenting a semi-porous zone | Wide rays | More or less distinct | Alternating parenchyma bands and fiber | *Cf. Ficus ou acacia* |
| Ho / PH / T. 226 | Diffuse pores, grouped from two to three pores and/or radially, sometimes with a semi-porous zone | Wide to very wide | More or less distinct | Expansion of the radii at the boundary of the ring | *Tamarix sp.* |
| Ho / PH / T. 255 | Diffuse pores with a semi-porous zone in the initial wood | Uniseriate or 2 seriate? | More or less distinct | Numerous xylophagous galleries | Undetermined |
| **Imported conifers** | | | | | |
| **Museum inv. num.** | **Presence of resin canals** | **Limit of tree-rings** | **Transition from initial to final wood** | **Particularity** | **Hypothesis species** |
| Ho / PH / T. 33 | Yes | Distinct | Abrupt transition from initial to final wood | Resin canals predominantly in the final wood and thin final wood | Cf. *Pinus* |
| Ho / PH / T. 43 | No | Distinct | Abrupt transition from initial to final wood | Thin final wood, sometimes tangential parenchyma | Cf. *Cupressus ou Cedrus* |
| Ho / PH / T. 85 | No | Distinct | Gradual transition from initial to final wood | Traumatic canals and tree-rings more or less thin | *Cedrus* sp. |
| Ho / PH / T. 106 | No | Distinct | Abrupt transition from initial to final wood | Very thin tree-rings, thin woody rays (3–5 cells) | *Juniperus* sp. |
| Ho / PH / T. 108 | No | Not always distinct | Gradual transition from initial to final wood | Traumatic canals | Undetermined |
| Ho / PH / T. 170 | Yes | Distinct | Abrupt transition from initial to final wood | Resin canals predominantly in the final wood and thin final wood | Cf. *Pinus* |
| Ho / PH / T. 172 | No | Distinct | Gradual transition from initial to final wood | Traumatic canals | *Cedrus* sp. |
| Ho / PH / T. 182 | Yes | Distinct | More or less abrupt transition between the initial and final wood | Resin canals predominantly in the final wood | Cf. *Pinus* |
| Ho / PH / T. 184 | Yes | Distinct | More or less abrupt transition between the initial and final wood | Resin canals predominantly in the final wood | Cf. *Pinus* |
| Ho / PH / T. 211 | Yes | Distinct | Abrupt transition from initial to final wood | Resin canals predominantly in the final wood | Cf. *Pinus* |
| Ho / PH / T. 225 | No | Distinct | Gradual transition from initial to final wood | Fast growing, very thin final wood | Undetermined |

ducts in their anatomy, but with traumatic tissues and a gradual transition from earlywood to latewood are probably cedar (*Cedrus* sp.). The anatomical markers of label HO43 are similar to cedar, but the sample also exhibits false rings and an abrupt transition between the early- and latewood, with the latter being very thin; we thus suppose that this label was made of cypress (*Cupressus sempervirens*). Finally, on label HO106, resin ducts and traumatic ducts are absent, the transition from earlywood to latewood is abrupt and the growth is very slow, it therefore probably was made from juniper (*Juniperus* sp.) wood.

## Toolmarks identified in XRCT images

Toolmarks, identified by positioning the labels at varying angles and different light orientations, were compared to those detected in the XRCT images. In general, the XRCT images

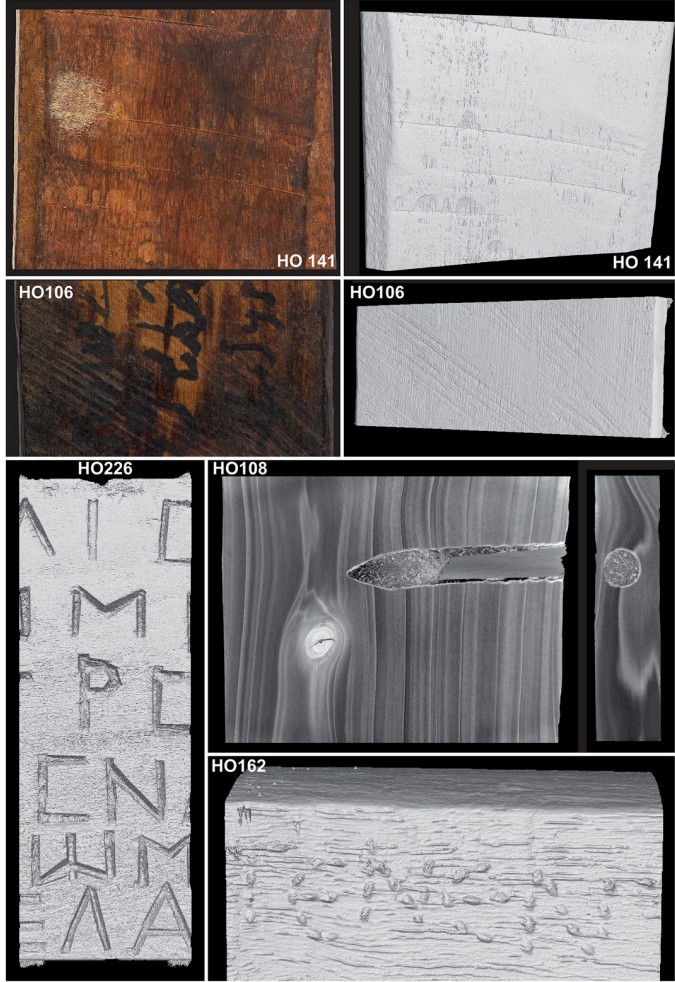

**Fig 6. Various toolmarks and features recognized on the mummy labels.**

validate the visual observations, but allow a finer interpretation on the orientation of the tool and penetration into the wood.

Several types of traces were regularly observed: sawing traces and marks left by sharp tools (i.e. axe, adze, plane and chisel or simply plane) (Fig 6, Table 3). The sawing marks are visible on the rough faces which normally does not contain inscriptions. Exceptions, however, such as the label HO106, which is inscribed on the roughly sawn face (Fig 6). A total of 21 labels shows traces of cutting tools, but the precise identification of the tool used remains complex. However, given the small size of many of the labels, plane-type tools seem to be ~~much~~ most suitable for flattening the face or faces intended to receive the inscription (Fig 6). Three other labels show both types of toolmarks (i.e. sawing and cutting tools). Eleven labels do not show any preserved toolmarks at all, probably because of wear or erosion of their surfaces or simply because of a completed finish.

Some arrangements, difficult to discriminate with the naked eye, are perfectly visible on the XRCT images. A hole with its peg still in place is, for example, found on label HO108 (Fig 6). The shape and undulations of the peg hole are clearly visible on the XRCT images, suggesting that it was drilled by a spoon drill. Finally, the XRCT images allow us to confirm that some

**Table 3. Description of the various data relating to the mode of cutting, traces of tools and the estimate size of the trees used (i.e. diameter, in cm).**

| Museum inventory number | Genus | Hypothesis species | Anatomical morphology | Cutting method | Tool marks | Type of inscription | Estimated diameter (in cm) |
|---|---|---|---|---|---|---|---|
| Ho / PH / T. 18 | Local hardwood? | Undetermined | Cambium | On a slab | cutting tools and saws | in ink | 4,8 |
| Ho / PH / T. 28 | Local hardwood? | *Tamarix sp.* | | On a slab | cutting tool | in ink | 18,6 |
| Ho / PH / T. 33 | Imported conifers | Cf. *Pinus* | | On a slab | cutting tool | in ink | suparallel |
| Ho / PH / T. 41 | Local hardwood | *Tamarix sp.* | | On a slab | saw | in ink | 14 |
| Ho / PH / T. 42 | Imported hardwood | *Salix sp.* | | On a slab | / | in ink | 7 |
| Ho / PH / T. 43 | Imported conifers | Cf. *Cupressus ou Cedrus* | | On mesh | / | in ink | 20 |
| Ho / PH / T. 47 | Local hardwood | *Ficus sp.* | | On a slab | / | in ink | 31,4 |
| Ho / PH / T. 58 | Local hardwood | *Ficus sp.* | | On a slab and a mesh | / | in ink | suparallel |
| Ho / PH / T. 61 | Local hardwood? | Undetermined | Bark | On a slab | cutting tool | in ink | 8,3 |
| Ho / PH / T. 65 | Imported hardwood? | Undetermined | Nearby pith | On a slab | cutting tool | in ink | 8 |
| Ho / PH / T. 68 | Imported hardwood? | Undetermined | | On a slab | cutting tool | in ink | 11,8 |
| Ho / PH / T. 69 | Local hardwood | *Tamarix sp.* | | On a slab | saw | in ink | 22,8 |
| Ho / PH / T. 85 | Imported conifers | *Cedrus* sp. | | On mesh | cutting tool | in ink | 27,8 |
| Ho / PH / T. 95 | Local hardwood | *Tamarix sp.* | | On mesh | cutting tool | in ink | suparallel |
| Ho / PH / T. 97 | Local hardwood | *Ficus sp.* | | On mesh | / | in ink | 13 |
| Ho / PH / T. 104 | Local hardwood | *Tamarix sp.* | | On a slab | saw | in ink | 17,8 |
| Ho / PH / T. 105 | Local hardwood | *Ziziphus sp.* | Pith | On a slab | cutting tool | in ink | 4,4 |
| Ho / PH / T. 106 | Imported conifers | *Juniperus* sp. | | On mesh | cutting tools and saws | in ink | suparallel |
| Ho / PH / T. 108 | Imported conifers | Undetermined | Pith | On a slab | cutting tool | in ink | 9,4 |
| Ho / PH / T. 111 | Local hardwood? | *Cf. Ficus ou acacia* | Pith | On a slab | cutting tool | in ink | 6,4 |
| Ho / PH / T. 112 | Local hardwood? | Undetermined | | On a slab | cutting tools and saws | in ink | 8,2 |
| Ho / PH / T. 117 | Local hardwood? | Undetermined | | On a slab | cutting tool | in ink | 10,4 |
| Ho / PH / T. 132 | Local hardwood | *Tamarix sp.* | | On a slab and a mesh | / | in ink | 13,4 |
| Ho / PH / T. 137 | Imported hardwood | *Fagus sp.* | | On a slab and a mesh | cutting tool | in ink | suparallel |
| Ho / PH / T. 138 | Imported hardwood | *Ulmus sp. Minor* | | On a slab | / | in ink | 19,6 |
| Ho / PH / T. 141 | Imported hardwood? | *Cf. salix* | | On a slab | cutting tool | in ink | 9,6 |
| Ho / PH / T. 162 | Local hardwood | *Cf. Ficus ou acacia* | Cambium | On a slab | cutting tool | engraved | 8,2 |
| Ho / PH / T. 170 | Imported conifers | Cf. *Pinus* | | On a slab | cutting tool | in ink | 21,7 |
| Ho / PH / T. 172 | Imported conifers | *Cedrus* sp. | | On a slab | cutting tool | in ink | 19,1 |
| Ho / PH / T. 182 | Imported conifers | Cf. *Pinus* | | On mesh | cutting tool | in ink | suparallel |
| Ho / PH / T. 184 | Imported conifers | Cf. *Pinus* | | On a slab and a mesh | cutting tool | in ink | 46,8 |

*(Continued)*

**Table 3.** (Continued)

| Museum inventory number | Genus | Hypothesis species | Anatomical morphology | Cutting method | Tool marks | Type of inscription | Estimated diameter (in cm) |
|---|---|---|---|---|---|---|---|
| Ho / PH / T. 199 | Imported hardwood | *Fagus sp.* | | On mesh | cutting tool | in ink | suparallel |
| Ho / PH / T. 210 | Imported hardwood | *Fagus sp.* | | On a slab | cutting tool | engraved | 31 |
| Ho / PH / T. 211 | Imported conifers | Cf. *Pinus* | | On mesh | / | engraved | 37,4 |
| Ho / PH / T. 225 | Imported conifers | Undetermined | | On a slab | / | engraved | 12,7 |
| Ho / PH / T. 226 | Local hardwood | *Tamarix sp.* | | On a slab | cutting tool | engraved | 22,6 |
| Ho / PH / T. 229 | Imported hardwood | *Fagus sp.* | | On a slab | / | in ink | 24,6 |
| Ho / PH / T. 255 | Local hardwood? | Undetermined | | On a slab | / | in ink | 8,6 |

inscriptions were carved into the wood with small chisels or punches. It was also possible to determine the number of chisel strokes required and the depth of the cuts (see, for example, labels HO162 and HO226 in Fig 6). Detailed analysis of the engraved inscriptions on label HO226 shows that all Greek letters, except for "pi" and "mu" (for which a 4 mm chisel was used) were made with a minimum of strokes using an 8 mm chisel (Fig 7). Some letters required at least two chisel strokes in the extension of the first one to form the longest strokes, such as the bars of "nu" or the sides of "alpha". The letters engraved in the center of the label and on the edges have a different depth (2.5 and 1 mm).

## Timber size estimation

Original wood sizes were estimated from XRCT images based on the convergence of woody rays and/or ring curvatures. In the case of three labels, HO18, HO61, and HO162, we could even take advantage of the presence of the cambium or bark to accurately determine the size of the tree used. For most of the labels, the outermost rings were absent, and the tree sizes presented here have to be considered a minimum estimate. Finally, as labels HO33, HO58, HO95, HO106, HO137, HO182, and HO199 were obtained from radial cuts, the low curvature of growth rings did not allow any estimation of tree size.

For the remaining 31 labels, the minimum tree diameters ranged from 44 to 468 mm (Table 3). Despite the small sample size, a distinction can be made between imported broadleaved, endemic broadleaved and imported conifer species. The sizes of imported conifers (94–468 mm, avg. 244 mm, n = 8) were significantly larger than those of imported broadleaves (70–310 mm, avg. 159 mm, n = 16) and especially of endemic broadleaves (44–314 mm, avg. 133 mm, n = 7) (Fig 8) These results must be qualified, however, by the method of cutting. Cutting on a slab from endemic broadleaves, imported broadleaves and conifers, labels had average diameters of 133, 159 and 157 mm respectively. The conifer labels made on mesh are from trees with larger diameters (average 284 mm).

## Conventional vs. XRCT dendrochronology

In general, tree-ring widths can be measured with different approaches depending on the visibility of growth rings. Measurements based on photographs are ideal if rings on the face or the flat side of mummy labels were wide and easily visible. In cases where tree rings were too narrow, the use of a measuring table was more appropriate. On many labels, neither of these two

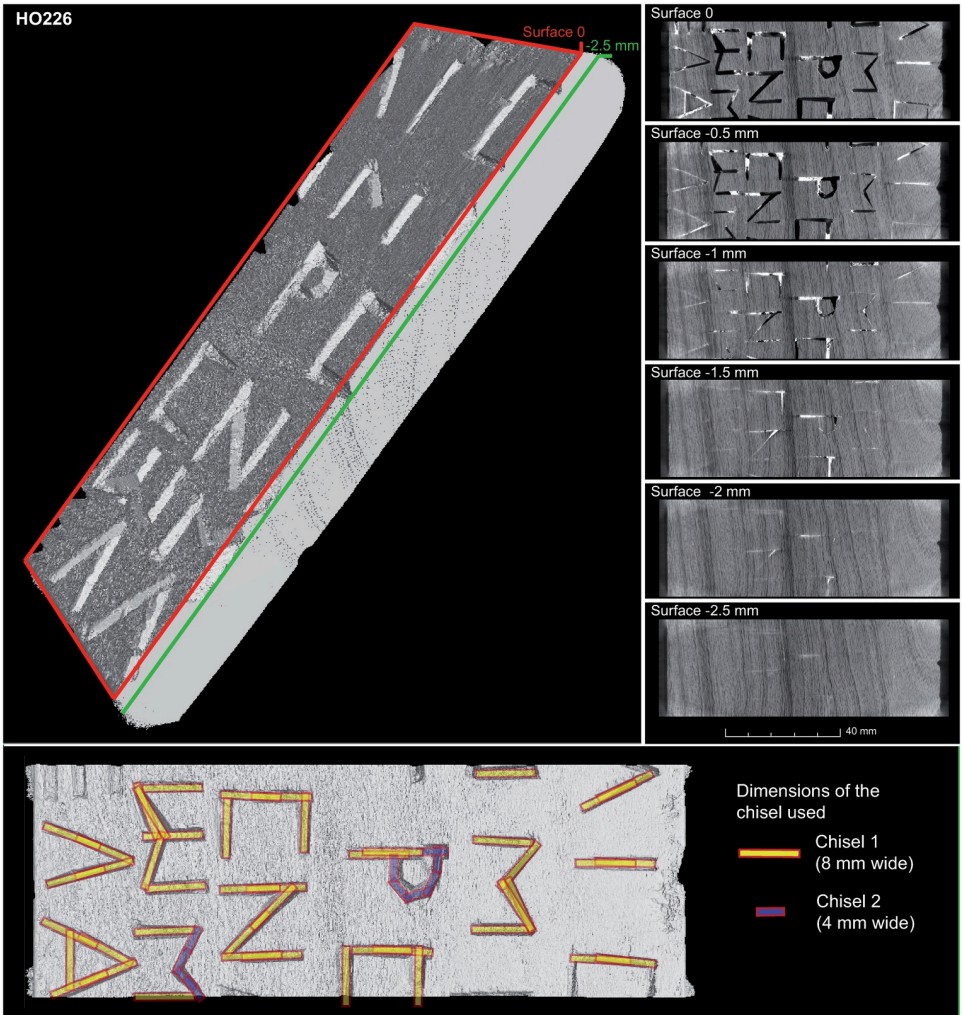

**Fig 7. Analyses of various chisel strokes and cutting depths as identified on a single label to make the inscription.** In addition, we can distinguish the use of at least two chisels with different widths.

approaches could be used easily due to (i) ring boundaries that were difficult to identify, (ii) rings that were too narrow (e.g., in the case of label HO106 for which average ring widths of 0.24 mm were recorded) or (ii) the presence of a coating (e.g., paint, wax, grease, patina of use) or inscriptions (engraved or inked) (see e.g. HO211) In these cases, XRCT images were the only non-destructive way to acquire and accurately delineate rings boundaries and to measure ring widths.

To analyze the advantages and disadvantages of the different methods, we compared ring-width measurements obtained from photographs or by using the measurement table with those measured on the transverse plane of XRCT images. As a rule of thumb, and with the exception of label HO182, the traditional measurements were superior to those made on XRCT images. Differences between methods, however, only rarely exceeded a few 0.01 mm and depended mainly on the cutting. In the case of radial ("on mesh") cutting (e.g. labels HO43, HO85, HO182 and HO184) close to the radial growth of rings, differences were very small (Fig 9). The discrepancy between the acquisition methods was maximal in those cases

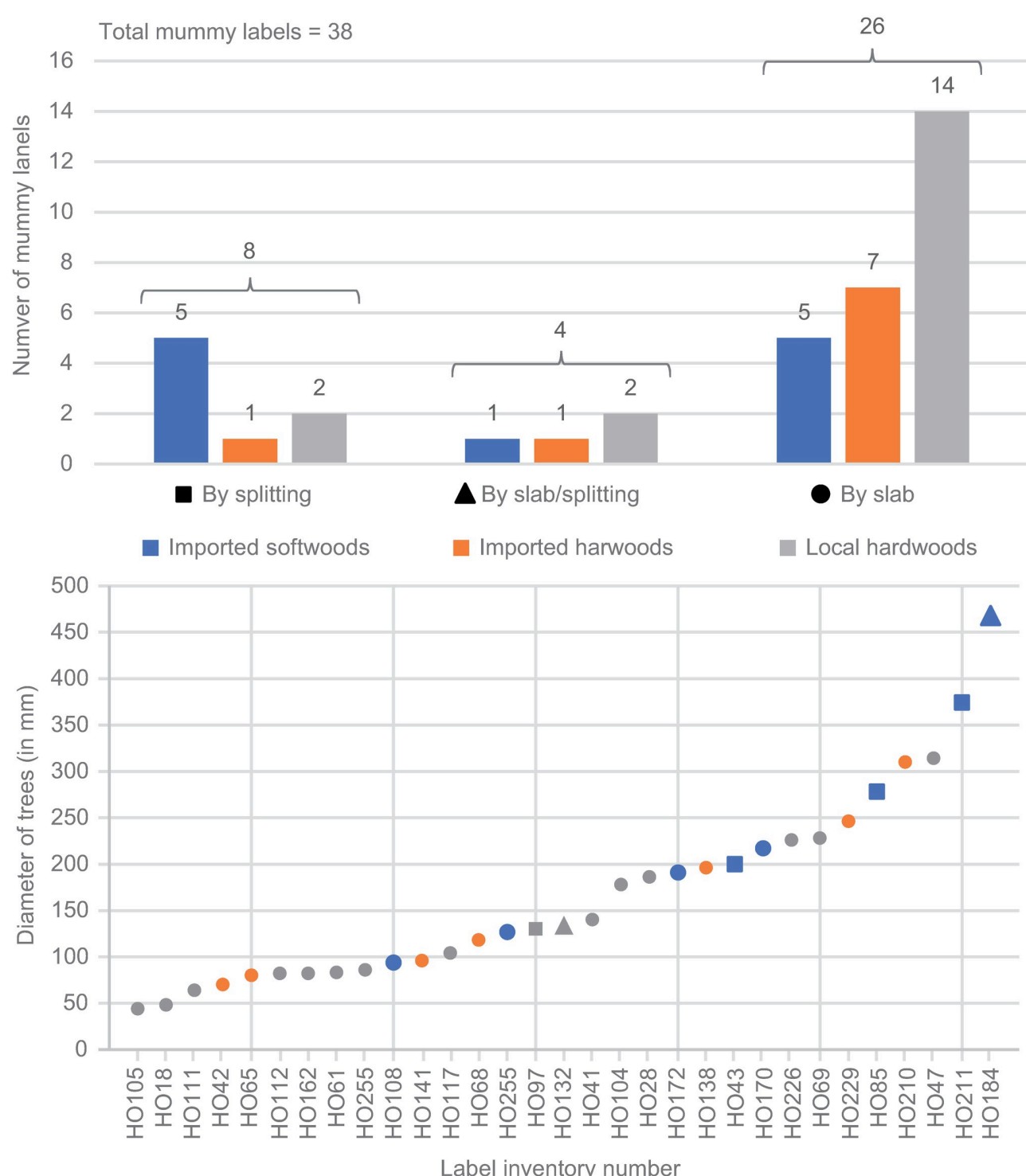

**Fig 8. Distribution of mummy labels by cutting method and tree family type, as well as estimated tree diameters used to produce the labels.**

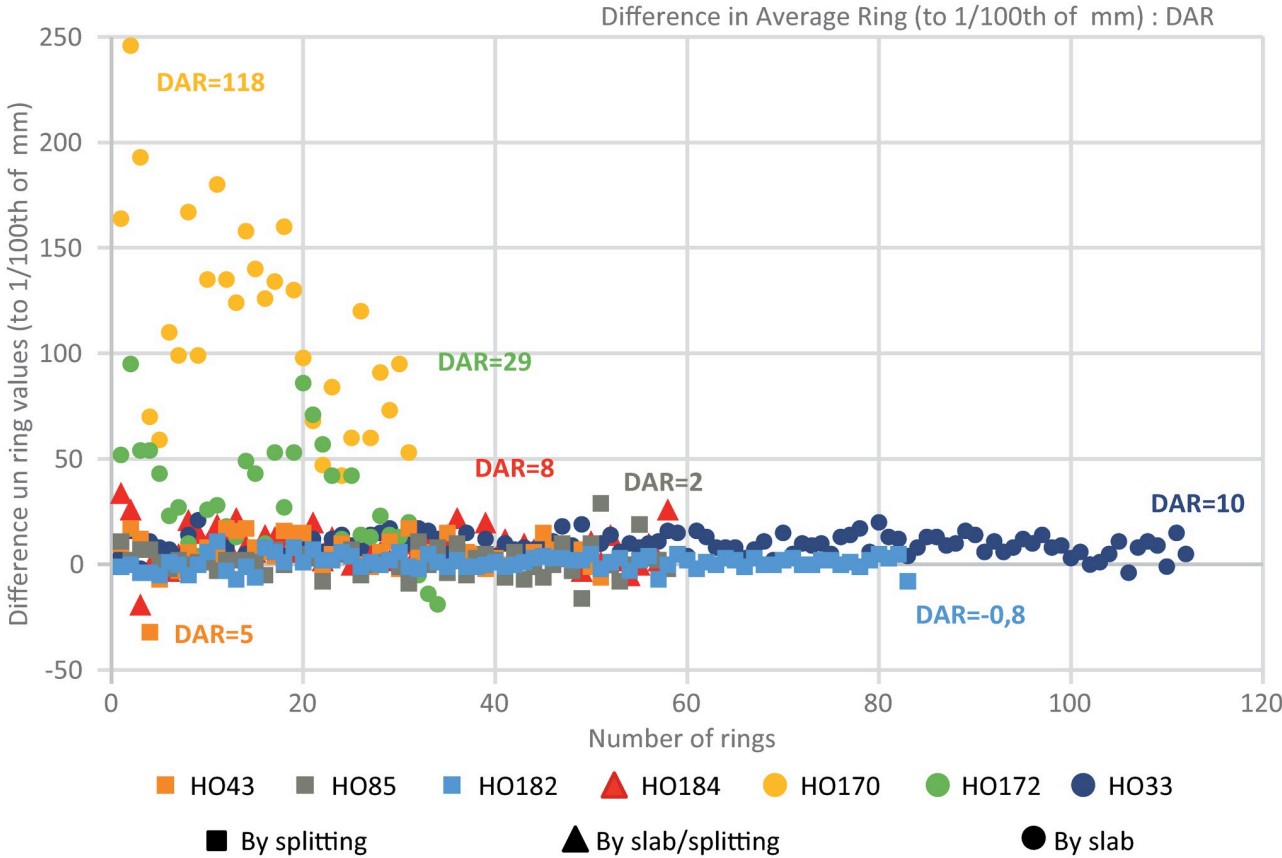

**Fig 9. Comparison of differences in ring-width values obtained from high-resolution photographs and CooRecorder, conventional measurements with a moving table and from XRCT images.**

where cutting was made on slabs or close to the pith (e.g., HO172 and HO170). In the case of HO170, the traditional methods exceeded on average by 117% the widths measured on XRCT images. However, this difference was heterogeneous and more important close to the pith or in the first growth rings, and also varied according to the orientation and curvature of the rings [8].

## Discussion

In this paper, 38 mummy labels were selected from the BNU Strasbourg collection and analyzed with X-ray computed tomography (XRCT) imagery. We evaluated the potential of XRCT images for an anatomical identification of wood species, the detection of toolmarks, assessment of original wood diameters and the measurement of ring widths and compared results with those obtained with traditional visual observations, optical (binocular, microscope) and digital (high resolution photography) analyses.

### Contribution of XRCT images to wood anatomy

XRCT images allows more precise insights into the morphology of certain anatomical markers of wood such as wood rays or the distribution, shape and number of pores, but also aid to better visualize ring boundaries. Yet, analyses are limited to the transverse plane because the

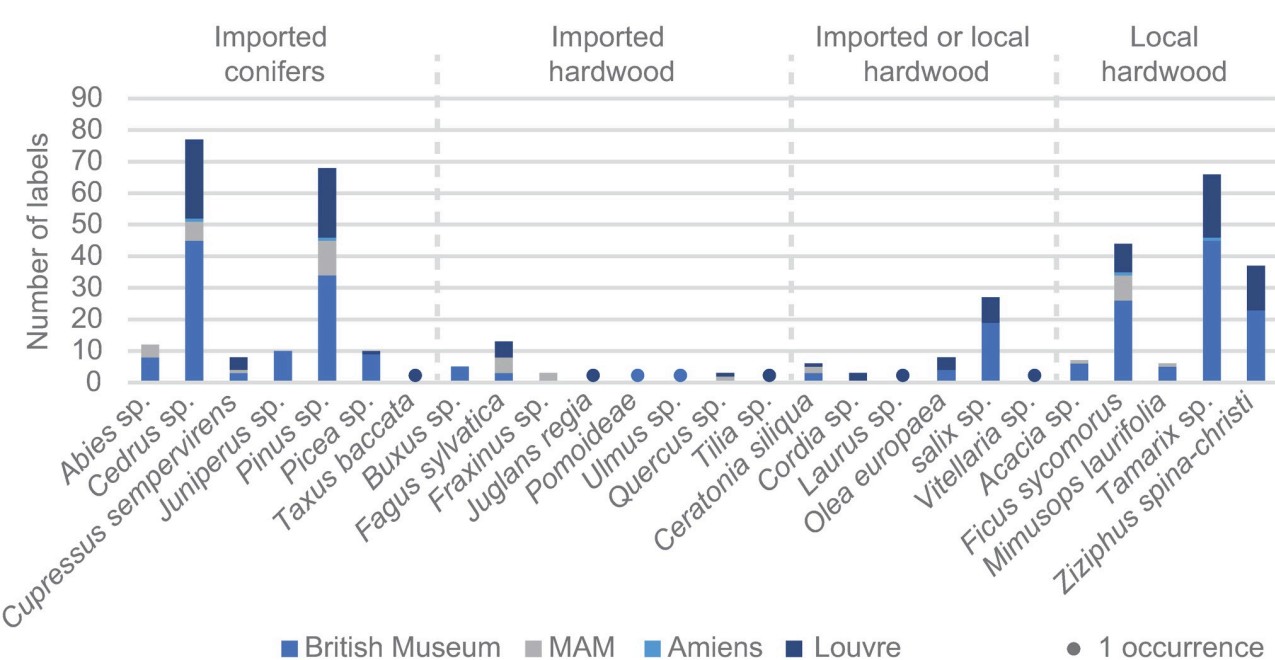

**Fig 10. Taxonomic variety of species identified in the mummy label collections of the British Museum, the Louvre, the Musée de Picardie in Amiens and the Musée Archéologique de Marseille.**

resolution of XRCT images is too low to properly assess anatomical markers on the tangential and radial planes. Magnifications of 100x to 400x would be required to facilitate recognition of anatomical markers [31], which is well beyond the current resolution of XRCT images. This limitation prevents formal identification of most species except for a few imported broadleaves with ring-porous zones (elm) or diffuse pores (beech, willow). For local species, XRCT images suggest that some labels are made of tamarisk, fig, acacia, or jujube, but a definitive identification will only be possible if one could measure the widths and heights of woody rays in the tangential and radial planes [33–34]. In the case of conifers, the presence of resin ducts and the transition from earlywood to latewood was used on XRCT images to differentiate species groups. Despite the undeniable benefits of tomography, species detection is not possible without the sampling of a small wood piece on the label [9].

A corpus of 420 mummy labels with identified species, from the collections of the British Museum (n = 250, https://www.britishmuseum.org/collection/), the Louvre (n = 122, https://collections.louvre.fr/; [42], and the Mediterranean Archaeology Museums of Marseilles (n = 44) and Amiens (n = 4) [29], supports some of our hypotheses (Fig 10). In particular, it confirms the existence of labels made of cedar (*Cedrus libani* or *atlantica*, n = 77, 41% of conifer labels) and pine (n = 68, 36%). Whereas cypress (n = 8, 4%) and juniper (n = 10, 5%) are much less represented, the proportions of species conserved in these museum collections seem to be consistent with our results (n = 1 for either of the species).

In the case of imported and local broadleaved species, a validation of our hypotheses is less obvious. Beech (*Fagus* sp.) is well represented in the collections (n = 13, or 46% of the imported broadleaves) as it is at the BNU in Strasbourg. The single elm (*Ulmus* sp.) label is attested by only one other occurrence in the British Museum collection (n = 1, 3.5%). In addition, the presence of a single row of pores in the well-characterized pore zone of label HO128 and wood rays that are 2–5 cells wide suggests the tree that was used to produce the label was a mounting elm (*Ulmus glabra*) [36].

In the case of the endemic species, fig (*Ficus sycomores*, n = 44, 27.5% of local broadleaves), tamarisk (*Tamarix* sp., n = 66, 41%) and jujube (*Ziziphus spina-christi*, n = 37, 23%) are well represented in our analyses (n = 3 for fig, n = 7 for tamarisk, n = 1 for jujube) and in the museum corpus containing 420 labels. On the other hand, we identified two acacia labels, while this species only represents 4% of the labels from local broadleaves, which may call our initial attribution into question.

Finally, willows (*Salix* sp.) are fairly well represented both in the museum collection (n = 27, 59% of imported and/or local broadleaved species) and in our analyses (n = 2). However, the complex identification of this diffuse-porous species in no way allows for a definitive attribution. Furthermore, the broad taxonomic spectrum [31, 34–35] includes more than ten different willow species for the two willow labels in the corpus. Some species, such as the Egyptian willow (*Salix subserrata*) identified on a mummy label from the Louvre [42] is endemic to Egypt, yet a distinction between imported and endemic deciduous is therefore impossible on the basis of XRCT images.

## Toolmarks on and tree sizes used for mummy labels

For a majority of the clear and visible toolmarks, the added value of XRCT as compared to observations under a binocular is relatively limited. In both cases, the distinction is focused to the traces of saws and cutting tools such as axes, adzes, planes, chisels and planes, all of which have been employed in ancient or Roman times [28, 36, 43, 44]. The interest of XRCT is, however, undeniable for the detection of faint or invisible toolmarks. In the case of the peg hole of label HO108 (Fig 6), for instance, tomography allows to confirm a recycling of the wood. That is, the pointed shape and undulations of the peg hole, clearly visible on the XRCT images, also suggest that it was drilled from a spoon bit, a tool known both in the Roman Empire [45, 46] and in ancient Egypt [47, 48]. Similarly, detailed analysis of the engraved inscriptions on label HO226 allows new insights into the gestures and practices of woodworking skills. The varying depths of the Greek letters, for example, may be related to a slight concavity of the label which ensures a good adhesion of the central part on the working surface but renders the chiseling more complex on the edges. The minimal number of chisel strokes also suggests that the inscription was made by a person (e.g., craftsman, scribe) with a perfect mastery of woodworking.

The added value of XRCT images in highlighting cutting methods and estimating sizes is especially important at the level of the ends of labels, which are often only poorly legible. Still underdeveloped [12, 14], this approach is crucial as it can help to better information on collection practices in a wood-importing country due to wood resources that are limited to the Mediterranean coast, the Nile Delta, the banks of the Nile and the main oases [49]. Despite the limitations of the small sample size, our analysis shows correlations between the cutting methods used, the size of trees and the species used to produce the mummy labels. These results show that the largest diameters estimated (on average 245.5 mm) are most often cut on mesh (probably by splitting) from conifers (n = 5). In the case of some labels (n = 7) with radial cutting, an estimation of the diameter was not possible because of the subparallel aspect of the wood rays and the weak curvature of rings, but in any case, these labels were cut from large caliber trees [38, 50]. Medium- to small-sized calibers (135 mm on average) are most often cut on a slab by sawing local broadleaved species. In the absence of traces corresponding to a length sawing, it is reasonable to think that the latter was carried out by means of a handsaw, known as a "pull saw", widely used in ancient Egypt [47] but not very adapted for large calibers. It should be noted, however, that reworking of wood, very common in Egypt [15], likely biases interpretations, as an initial cutting on a mesh can be reworked later with a saw, especially in

the case of dry wood. Finally, the frequent use of saws on local woods indicates that these woods, which are gnarled and very dense due to rapid and continuous growth, are not suitable for cutting by splitting due to the lack of a preferential splitting plane. Different craft practices are therefore used depending on the origin of the wood exploited and the tools used.

### Ring-width measurements based on different approaches and the dendrochronological potential of species endemic to Egypt

The measurement of ring widths was realized on images, with classical ring-width measurements on a table on the flat side of the mummy labels and by using XRCT images. Significant differences exist in the quality of results, depending on the cutting mode used. These differences are largest in the case of cutting on a slab close to the pith (HO170 and 172). Despite the small size of our sample, we hypothesize that dendrochronological analyses realized previously on the flat side of other mummy label collections may thus potentially be biased. However, the absence of a reference for the eastern Mediterranean basin in Antiquity does not allow quantification of the impact of these biases on sample dating. Regarding inter-series synchronization, only labels HO182 and HO184 show Student's T values >5 and r correlation values >0.7 for an overlap of 28 years, suggesting the use of contemporary woods. For these two labels, correlations obtained between the ring-width series measured on XRCT images (r = 0.764, T = 5.3) are higher than those obtained between XRCT and traditional tree-ring measurements (r = 0.717 and 0.763, T = 5.151 and 4.824) or between traditional dendrochronological measurements (r = 0.715, T = 4.691). These results obviously need to be replicated but suggest an added value of XRCT for the measurement of tree-ring widths as well as the synchronization and dating of series, most certainly due to perfectly oriented measurements along the transverse plane.

XRCT images also confirm the limited dendrochronological potential of endemic species. For the construction of dendrochronological time series, a sufficiently large number of distinct, annual rings is needed for different series to be synchronized with each other [51]. None of the species identified with XRCT images appear to meet these requirements for the time being. XRCT analyses did neither allow identification of ring boundaries in sycomore (*Ficus sycomorus*) or common (*Ficus carira*) fig, nor in the case of the acacia labels. Consistent with observations by Kuniholm *et al.* [17], Creasman [13] and El Sherbiny [52], the parenchyma bands visible in the XRTC images do not, unfortunately, correspond to ring boundaries. In the case of fig, this is confirmed by the distribution of pores which are either isolated or radially abutting, but mostly distributed between fiber and parenchyma bands. The seven labels potentially made of tamarisk and the label produced from jujube sometimes show distinct rings, as described by Gale & Cutler [33], but the legibility of rings is very heterogeneous and the number of rings is often below 20. This is in line with observations made on living trees in Africa and the Near East on several species, including acacia and tamarisk [53–61], showing rapid growth and short series that are difficult to use for the elaboration of dendrochronological references. Therefore, imported conifers and broadleaved species remain the best candidates for the construction of reference chronologies for this part of the Mediterranean [17, 19, 62–64].

### Conclusions

The results obtained from 38 mummy labels from the BNU in Strasbourg confirm the potential of X-ray computed tomography (XRCT) for the analysis of archaeological material [9]. Even if species could not be formally identified due to the insufficient resolution of the XRCT images on the longitudinal planes [9]. The results obtained on the transverse planes show that

11 labels are probably from imported conifer, 10 from imported broadleaved and 17 from local broadleaved species.

XRCT is not indispensable for the recognition of toolmarks. It does, however, undeniably improve observations of the number and depth of tool blows, highlighting developments invisible to the naked eye and providing an unambiguous view on cutting methods. It constitutes, therefore, an innovative approach to provide new information on the silvicultural practices and skills of Egyptian craftsmen.

As far as the dendrochronological approach is concerned, XRCT, by its non-destructive character, appears particularly adapted to the analysis of museum collections. XRCT images of the transverse plane offer a clear view of possible ring boundaries [7–9] that are often difficult to observe and measure using traditional methods. XRCT therefore allows insights into objects and exploration of the full potential of archaeological materials for dendrochronological purposes. Highly-resolved XRCT images can thereby also contribute to the elaboration of solid reference chronologies that can then be used for dating purposes or the analysis of past climates.

In terms of accuracy, measurements made on the flat side of conifer labels from traditional approaches and on the transverse plane of XRCT images do not show significant differences in the case of radial cutting of labels. If, however, labels were cut on a slab, differences are sometimes significant near the pith or for the first rings depending on the orientation of the rings, with likely impacts on the synchronization of ring widths. It would be relevant to develop approaches further to correct these differences for measurements made on the flat side of objects.

In terms of potential, XRCT images confirm the limited dendrochronological potential of several local species (fig, tamarisk, acacia, jujube) as they do not show distinct rings, rarely visible rings, or only a small number of rings ($<20$) insufficient for cross-dating and the construction of robust reference chronologies. Although other approaches could be tested (such as quantitative wood anatomy) to increase the dendrochronological potential of these local species, our results suggest that future work should focus on imported conifer labels, which are easier to measure radially and likely to provide longer series [15].

## Acknowledgments

We warmly thank the engineers of the Institut Charles Sadron, CNRS—UPR22 of Strasbourg, Damien Favier and Antoine Egele, for their professionalism and for taking the time to process all the selected mummy labels. We would like to thank Victoria Asensi Amorós for her advice concerning anatomical identifications and the exploitation of tomographic images.

## Author Contributions

**Conceptualization:** François Blondel.

**Data curation:** François Blondel, Gisela Bélot.

**Formal analysis:** François Blondel.

**Funding acquisition:** François Blondel.

**Investigation:** François Blondel.

**Methodology:** François Blondel.

**Resources:** François Blondel, Gisela Bélot.

**Supervision:** Sabine R. Huebner, Markus Stoffel.

**Writing – original draft:** François Blondel, Gisela Bélot.

**Writing – review & editing:** Christophe Corona, Sabine R. Huebner, Markus Stoffel.

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
