## [Decision Letter · Decision Letter 0]

3 Oct 2023

PONE-D-23-11489The potential of X-ray computed tomography for a xylological and dendrochronological analysis of Egyptian mummy labelsPLOS ONE

Dear Dr. Blondel,

Thank you for submitting your manuscript to PLOS ONE. After careful consideration, we feel that it has merit but does not fully meet PLOS ONE’s publication criteria as it currently stands. Therefore, we invite you to submit a revised version of the manuscript that addresses the points raised during the review process.

We look forward to receiving your revised manuscript.

Kind regards,

Dario Piombino-Mascali, Ph.D.

Academic Editor

PLOS ONE

“Yes.

This article is part of a post-doctoral project funded by the SNSF, project n°192176 "The Roman Egypt Laboratory: Climat Change, Societal Transformations, and the Transition to Late Antiquity"

web page project: https://ancientclimate.philhist.unibas.ch/en/project/

Web page SNSF :

https://www.snf.ch/fr”

“The tomographic analyses carried out on 38 mummy labels was financed in the framework of the SNSF project n° 192176 "The Roman Egypt Laboratory: Climate Change, Societal Transformations, and the Transition to Late Antiquity". We warmly thank the engineers of the Institut Charles Sadron, CNRS - UPR22 of Strasbourg, Damien Favier and Antoine Egele, for their professionalism and for taking the time to process all the selected mummy labels. We would like to thank Véronique Asensi Amoros for her advice concerning anatomical identifications and the exploitation of tomographic images.”

“Yes.

This article is part of a post-doctoral project funded by the SNSF, project n°192176 "The Roman Egypt Laboratory: Climat Change, Societal Transformations, and the Transition to Late Antiquity"

web page project: https://ancientclimate.philhist.unibas.ch/en/project/

Web page SNSF :

https://www.snf.ch/fr”

4. We note that Figures 1, 2, 4, 6 and 7 in your submission contain copyrighted images. All PLOS content is published under the Creative Commons Attribution License (CC BY 4.0), which means that the manuscript, images, and Supporting Information files will be freely available online, and any third party is permitted to access, download, copy, distribute, and use these materials in any way, even commercially, with proper attribution. For more information, see our copyright guidelines: http://journals.plos.org/plosone/s/licenses-and-copyright.

1. You may seek permission from the original copyright holder of Figures 1, 2, 4, 6 and 7 to publish the content specifically under the CC BY 4.0 license.

6. We note that you have referenced (John J. Variation of wood anatomy in relation to environmental factors in two southern African broadleaves. Departement of Pure and Applied Biology, imperial College, London, Unpublished PhD Thesis; 1990) which has currently not yet been accepted for publication. Please remove this from your References and amend this to state in the body of your manuscript: (ie “Bewick et al. [Unpublished]”) as detailed online in our guide for authors http://journals.plos.org/plosone/s/submission-guidelines#loc-reference-style

Additional Editor Comments:

Please consider the recommendations of the reviewers, make the relevant changes, double check the reference style and have the paper edited by a native English speaker prior to resubmission. Thank you

Reviewers' comments:

Reviewer's Responses to Questions

**Comments to the Author**

1. Is the manuscript technically sound, and do the data support the conclusions?

Reviewer #1: Yes

Reviewer #2: Yes

Reviewer #3: Yes

2. Has the statistical analysis been performed appropriately and rigorously? 

Reviewer #1: N/A

Reviewer #2: N/A

Reviewer #3: N/A

3. Have the authors made all data underlying the findings in their manuscript fully available?

Reviewer #1: Yes

Reviewer #2: Yes

Reviewer #3: No

4. Is the manuscript presented in an intelligible fashion and written in standard English?

Reviewer #1: Yes

Reviewer #2: Yes

Reviewer #3: Yes

5. Review Comments to the Author

Reviewer #1: Nice piece of work. A few changes in terms of grammar and spelling (on mss that will be sent to the authors). A question: at the outset, how did the authors establish the type of wood, or was that a result of their work? A trifle unclear. Please clarify.

Reviewer #2: Dendrochronology remains underdeveloped in the context of Egyptian archaeology; therefore, this paper makes a significant contribution to the field of 'dendro-Egyptology.' However, the applicability of dendrochronology in ancient Egypt is hindered by the scarcity of trees with suitable rings for dating purposes in the region. Dendrochronology is most reliable when there is a continuous sequence of tree rings spanning a long period. In regions with limited tree growth, such as Egypt, this method becomes less practical due to the rarity of suitable tree species and samples. A primary challenge in 'dendro-Egyptology' is the limited availability of local wood resources, with wood reuse being a common practice. As mentioned in the paper, some of the examined labels originated from reused wood, which complicates dendrochronological analysis, not only here but in general in Egypt. Emphasizing this obstacle in the text is essential.

Nonetheless, mummy labels and wooden tablets with inscriptions bearing dates are of utmost importance for advancing dendrochronology in the region. Some wood identification methods can be invasive, and museum curators may be reluctant to employ them. Accurate species identification is crucial for dendrochronological examination, but it is often a challenging and sometimes impossible task. Therefore, this paper describes the available methodology, which may prove valuable to other scholars.

Over all I do not have negative comments and I do not see and flaws but regarding Table 1: it is stated that the dimensions are provided in millimetres. I believe there may be an error because it seems unlikely that these labels measure, for example, 14.5 mm in length; rather, it is more likely that they are 14.5 cm.

In other aspects, the paper appears sound. The discussion is clear, and the conclusion is acceptable.

Despite the limitations of the currently available equipment, studies like this one must be conducted as they hold the potential to refine our understanding of silvicultural practices, chronology, and local craftsmanship. The present study shows new possibilities for the application of XRCT while also highlighting its limitations. Negative results are equally valuable, and the authors provide valuable directions for further research, which should be pursued. Therefore, this paper provide a valuable contribution.

Beyond the aim of the paper it is worth mentioning that the methodology described has the potential to be applied to other projects examining wooden objects from various periods and regions. For instance, at the last International Congress of Egyptologists, T. Beck presented a very interesting talk titled 'From Style to Function: Wooden statues and their ritual entanglements,' revealing a previously unknown practice of creating depictions located in concealed parts of wooden statues. Applying the methodology presented here to search for such engraved depictions in joints of wooden objects could be applied in future research.

Reviewer #3: Dear authors,

you present a very interesting study about testing and illustrating the potential of X-ray computed tomography (XRCT) to investigate the production of mummy labels from the Roman Era. The focus of the research is to assess whether XRCT can be used to i) identify the wood species, ii) retrieve tool traces and iii) carry out non-invasive dendrochronology. Your manuscript is very well structured and clearly written and you highlight the limitations of the technique, which is a crucial point in this type of papers, and one very much appreciated by this reviewer. Having said that, and as much as I would like to see this article published swiftly, I wonder whether it is suited for PLOS ONE. I base this thought in that i) it deals with the implementation of a technique that is not novel to cultural heritage research in general (the novelty resides in the object to which it is applied to), and ii) it does not report major findings. Given its informative character about the implementation of a technique, it may be better suited for a journal such as Heritage Science, or Journal of Archaeological Science: Reports.

In any case, since the article presents sound research, I will have recommended minor revision, and leave the decision to the editor. Should the editor accept it, I have some minor suggestions for improvements that you may want to address (also if you send it elsewhere):

- In the submission file you replied to the Data Availability statement that "Yes - all data are fully available without restriction". However, you have not indicated where this data can be found. This statement should be amended providing a link to the data repository.

- Replace non-destructive by non-invasive in the abstract and elsewhere, as a non-destructive intervention can still be invasive, but the work you describe here is non-invasive (which by definition is also non-destructive).

- In the abstract, mention the nr of labels that were examined, also in the introduction, as this is not explicitly said until the first paragraph of the discussion (38 labels). The number of researched labels contrasts with the one reported in the introduction for the collection in question (256 labels). “Here, XRCT was tested on a batch of 38 mummy labels from Roman Egypt”

- The references 1 to 5 in the introduction, and many other throughout the text, are biased towards French publications that are difficult (or impossible) to obtain online. E.g: 10, 11, 14, 24, 26, 29, 32, 39, 42 (there are several English publications to refer to the principles of Dendrochronology). References 43 and 44 are highly justified for this article, but for the other ones, nowadays there is a large corpus of scientific literature in English dealing with the research of wooden object from museum collections in Austria, Italy, The Netherlands, Norway, etc. and about other topics related to this paper. PLOS ONE is a multidisciplinary journal read worldwide and providing English references wherever possible is an added value. I suggest that you replace those references as much as possible by scientific literature published in English.

- In the first paragraph of Material and Methods, please write what dates correspond to the Greco-Roman period, as this is not known by readers not familiar with Egyptian history.

- Also in the Methods, in the part about X-ray CT, please specify the X-ray settings used, as this is customary for X-rau CT studies, and is necessary for reproducibility of results. Also, please explain whether the same settings were used for labels of different dimensions (did you adjust the settings or used the same for all).

- Figure 1 is the same one as the one published in the article https://doi.org/10.1163/27723194-bja10017 by the same authors earlier this year: ‘Mummy Labels: A Witness to the Use and Processing of Wood in Roman Egypt’. I see that the article is published in the IJWC under license CC BY 4.0 Deed, therefore it is up to PLOS ONE whether they are OK with the figure being published there too.

- In the last paragraph of the point "Comparison of dendrochronological acquisition methods", you report a find that echoes what Bossema et al 2021 published too: « The discrepancy between the acquisition methods was maximal for cutting on slabs or close to the pith" (they show a figure of tree-ring measurements obtained from the board of a historical chest that clearly illustrates that discrepancy). Therefore I suggest that you refer here to their publication as well, as it will show that you are well acquainted with the current literature (you already have their publication in the reference list).

- In line 254 there is no need to use the acronym QWA, as this is not used later on.

Kind regards

6. PLOS authors have the option to publish the peer review history of their article (what does this mean?). If published, this will include your full peer review and any attached files.

Reviewer #1: No

Reviewer #2: **Yes: **Wojciech Ejsmond

Reviewer #3: No

---

## [Author Response · Author response to Decision Letter 0]

16 Jan 2024

Response to editor

Thank you for your comments on our revised article. We have made stylistic changes in line with the templates. We have added the financial information requested on the site and removed this information from the acknowledgements and placed it in the dedicated section. As you indicated, I have also modified the cover letter to read: This article is part of an SNSF-funded post-doctoral project, project n°192176 "The Roman Egypt Laboratory: 

Climat Change, Societal Transformations, and the Transition to Late Antiquity" (Project website: https://ancientclimate.philhist.unibas.ch/en/project/ ; SNSF web page :

https://www.snf.ch/fr). The tomographic acquisitions of the 38 mummy labels were funded as part of SNSF project n°192176 "The Roman Egypt Laboratory. I also made it clear that the SNSF (the funders) did not influence the writing of the article and the results that followed from it.

Our intention was for the figures to be copyright free. This is simply an oversight on our part during our first submission, since all the figures were made for the article, with the exception of figure 1 already used in a previous article, but which is already in free access. They are therefore not subject to copyright. This error has now been corrected. We therefore authorize the publication of all figures under CC by 4.0 license. In detail, I would like to point out that Figure 1 is a photograph taken by S. R. Huebner, co-author of the article. Figure 2 is a composition based on a photograph of a label, which I took myself during the tomographic sessions in Strasbourg, and on the detail of a label from the BNU collection (which is one of the photographs from the collection of the BNU deposited on the zenodo server linked to the article). Figure 3 is a figure that I created from the extraction of a tomographic image that I made from the dataset available on the zenodo server as part of the article. Figure 4 was produced from a front and back montage of the HO184 label (available on the server) and the extraction of views from the tomographic dataset that I produced as well as the growth curves of my dendrochronological measurements. Figures 5, 6 and 7 are a tomographic imaging montage that I created from the entire tomographic image data and photographs of the tool marks that I made myself. Figures 8, 9 and 10 consist of graphs from the anatomical, dendrological and dendrochronological observations that I made.

All the tomographic images of the 38 labels and the photographs of the latter taken by the BNU of Strasbourg, but in open access (public data) and a table summarising the tomograph settings for each label, as well as the views used to make the ring width measurements, have been archived as a compressed file and made available on a public repository (Zenodo). A text detailing the approach and protocol has been drafted and will be submitted to you. Data available on Zenodo will be made available upon acceptance of this article. 

We do not understand that the reference "John J. Variation of wood anatomy in relation to environmental factors in two southern African broadleaves. Department of Pure and Applied Biology, imperial College, London, Unpublished PhD Thesis; 1990" is not acceptable, even though it is considered as such and is accessible on the imperial college of london website. But to be on the safe side, we have marked it as unpublished in the text and highlighted it in the bibliography, based on your comments. We have written a response to each of the reviewers. Corrections taken into account by the reviewers are highlighted in green. Some additional corrections have been made by the authors and are shown in purple in the commented text version.

Response to Reviewers

Reviewer 1

Thank you for your proofreading and the grammar and spelling corrections. Everything has been taken into account. We have specified how we selected the mummy labels according to their taxonomic diversity on the basis of initial observations under binoculars. This detail has been included in the "selection of tomographed labels" section.

Reviewer 2

We would like to thank you for your review and for your comments on our article. We have added a sentence to the introduction to point out the difficulty of establishing dendrochronological references for Egypt in general, and for the Roman period in particular, due to the practice of reusing wood. We have also modified tables 1 and 3, where the dimensions are in cm rather than mm. It was indeed a mistake on my part.

Reviewer 3

Thank you for your review and your many comments and remarks. You are right about the unrestricted availability of the data, and this is also a request from the SNSF. All the tomographic images have been deposited on a dedicated server so that they are accessible and the results reproducible. We have replaced "non-invasive" with "non-destructive" throughout the article. We have corrected and indicated the number of labels analysed in the summary and specified in the introduction between those taken into account in this study and the collection as a whole. Most of the French references have been replaced by English references, but some remain difficult to replace. I am thinking in particular of Catherine Lavier's work on archaeometric approaches to wooden collections from archaeology or museum collections, as this is not a case study but a global approach to possible approaches to these collections, or Béal Arnold's work, which is a mass of data and information on dendrochronological approaches to pirogues discovered in Europe, which also has no equivalent in English. We specified the Greco-Roman period as requested. Indeed, we had not given all the scan settings. All these details have been added to the method, along with confirmation that the settings were the same for all 38 labels. It is true that Figure 1 is the same as the article published in the IJWC, but PlosOne does not seem to have objected to including it in this new article. The reference to Bossema et al. 2021 has been added as requested. Finally, we have removed the acronym QWA in the conclusion.

---

## [Editor Report · Decision Letter 1]

9 Feb 2024

PONE-D-23-11489R1The potential of X-ray computed tomography for a xylological and dendrochronological analysis of Egyptian mummy labelsPLOS ONE

Dear Dr. Blondel,

Thank you for submitting your manuscript to PLOS ONE. After careful consideration, we feel that it has merit but does not fully meet PLOS ONE’s publication criteria as it currently stands. Therefore, we invite you to submit a revised version of the manuscript that addresses the points raised during the review process.

**Please have the paper read by a native English speaker and adjust the reference style**

We look forward to receiving your revised manuscript.

Kind regards,

Dario Piombino-Mascali, Ph.D.

Academic Editor

PLOS ONE

Journal Requirements:

**Additional Editor Comments:**

Please adjust the references according to the journal style and have the paper read by a Native English speaker one last time prior to resubmission. I will accept the paper immediately afterwards.

---

## [Author Response · Author response to Decision Letter 1]

20 Apr 2024

Dear Plos One editors,

I have taken into account all suggestions from previous revisions proposed by Plos One reviewers and editors. This new version of the manuscript concerns changes made by a native English speaker who over-keyed words and changed several turns of phrase. These modifications concern only the synthaxis. I have updated and checked the bibliographical references. In this new submission, I'm submitting a new version of the manuscript with tracked corrections and another just corrected as requested. I've added a paragraph in the "letter rebuttal" which presents these modifications along with all those already presented in my previous submissions.

Concerning the data used and the possibility of replicating results, I have deposited all my data used for this article on two zenodo links with a protocol. This solution seemed simpler than the one proposed by "in protocols.io", especially for tomographic imaging data.

I hope that this submission will comply fully with PlosOne's requirements.

---

## [Editor Report · Decision Letter 2]

30 Apr 2024

The potential of X-ray computed tomography for a xylological and dendrochronological analysis of Egyptian mummy labels

PONE-D-23-11489R2

Dear Dr. Blondel,

We’re pleased to inform you that your manuscript has been judged scientifically suitable for publication and will be formally accepted for publication once it meets all outstanding technical requirements.

Kind regards,

Dario Piombino-Mascali, Ph.D.

Academic Editor

PLOS ONE

Additional Editor Comments (optional):

thanks for all your hard work
---

## [Editor Report · Acceptance letter]

29 May 2024

PONE-D-23-11489R2 

PLOS ONE

Dear Dr. Blondel, 

I'm pleased to inform you that your manuscript has been deemed suitable for publication in PLOS ONE. Congratulations! Your manuscript is now being handed over to our production team.

Kind regards, 

on behalf of

Dr. Dario Piombino-Mascali 

Academic Editor

PLOS ONE